# Optimal Use of Bispecific Antibodies for the Treatment of Diffuse Large B-Cell Lymphoma in Canada

**DOI:** 10.3390/curroncol32030142

**Published:** 2025-02-28

**Authors:** Isabelle Fleury, David MacDonald, Mona Shafey, Anna Christofides, Laurie H. Sehn

**Affiliations:** 1Institut Universitaire d’Hémato-Oncologie et de Thérapie Cellulaire, University of Montreal, Montreal, QC H3T 1J4, Canada; 2Division of Hematology, University of Ottawa, Ottawa, ON K1N 6N5, Canada; davidmacdonald@toh.ca; 3Department of Medicine, University of Calgary, Calgary, AB T2N 1N4, Canada; 4IMPACT Medicom Inc., Toronto, ON M6S 3K2, Canada; anna@impactmedicom.com; 5BC Cancer Centre for Lymphoid Cancer, The University of British Columbia, Vancouver, BC V6T 1Z2, Canada

**Keywords:** CAR-T therapy, bispecific antibodies, glofitamab, epcoritamab, diffuse large B-cell lymphoma, relapsed/refractory lymphoma

## Abstract

CAR-T cell therapy has significantly improved outcomes for patients with relapsed or refractory (R/R) diffuse large B-cell lymphoma (DLBCL), but challenges such as limited resources, manufacturing timelines, and notable toxicities persist. Bispecific antibodies (BsAbs), including glofitamab and epcoritamab, have demonstrated promising efficacy and represent a new treatment option in patients who are unsuitable for or have relapsed following CAR-T therapy. Bispecific antibodies have a manageable safety profile and are generally more widely accessible than CAR-T cell therapy. Case discussions in this paper illustrate the potential real-world application of BsAbs, highlighting their role in treating patients who have relapsed after or are unable to undergo CAR-T cell therapy. Overall, glofitamab and epcoritamab represent valuable treatment options in the evolving landscape of R/R DLBCL.

## 1. Background

In North America, diffuse large B-cell lymphoma (DLBCL) is the most common aggressive subtype of non-Hodgkin lymphoma (NHL) [1,2]. R-CHOP (rituximab, cyclophosphamide, doxorubicin, vincristine, and prednisone) remains the standard first-line treatment in Canada, with survival rates almost similar to the general population when patients are disease-free for at least two years post-therapy [3]. However, 35% to 40% of patients are refractory or relapse (R/R) after R-CHOP [3]. Historically, platinum-based salvage chemotherapy followed by high-dose chemotherapy and autologous stem cell transplantation (HDT-ASCT) offered curative potential in approximately one-quarter of eligible patients [4], although patients with primary refractory disease or who experience early relapse (within one year of treatment) have a particularly poor prognosis [5].

CAR-T cell therapy has emerged as a superior option to platinum-based salvage chemotherapy and HDT-ASCT for patients with refractory or early relapsing disease, with axicabtagene ciloleucel and lisocabtagene maraleucel demonstrating significant reductions in both relapse risk and mortality in the second-line setting [6,7]. Axicabtagene ciloleucel is now the standard of care in Canada for patients with refractory disease or who relapse within one year of completion of front-line treatment, with lisocabtagene maraleucel not funded yet in Canada [8]. The CAR-T cell therapies axicabtagene ciloleucel [9], lisocabtagene maraleucel [10], and tisagenlecleucel [11] have also shown promising efficacy in the third-line setting and represent the preferred treatment option in Canada for suitable patients who have not previously received it in the second-line setting [8]. However, CAR-T cell therapy may be associated with notable toxicities, such as cytokine release syndrome (CRS) and immune effector cell-associated neurotoxicity syndrome (ICANS), as well as severe and prolonged cytopenias [12,13].

Access to CAR-T cell therapy remains a challenge due to logistical, geographical, and resource-related constraints [3]. Furthermore, eligibility for CAR-T cell therapy is often restricted due to patient comorbidities, potential CAR-T cell-related toxicities, and rapidly progressing disease that may not allow adequate time for CAR-T cell manufacturing [14]. Up to one-third of patients referred for CAR-T cell therapy across Canada do not proceed, primarily due to prohibitive lymphoma progression [15]. In patients eligible for CAR-T cell therapy in the United States, around 10% [12] to 25% [13] of those at academic centers and around 50% of those in community settings [16] do not proceed, mainly due, again, to disease progression or a decline in clinical status. Moreover, of those undergoing leukapheresis, around 10% of patients do not undergo CAR-T cell infusion because of disease progression, a decline in clinical status, manufacturing issues, or other reasons [16,17]. Post-CAR-T cell therapy relapse represents an additional challenge, as survival is poor with conventional therapies [18]. Therefore, accessible, effective, and well-tolerated novel treatments are needed for patients unsuitable for, unable to receive, or progressing in terms of disease following CAR-T cell therapy.

Bispecific antibodies (BsAbs) comprise a class of engineered antibody products designed to simultaneously target two different antigens [1]. Epcoritamab and glofitamab are two BsAbs approved by Health Canada. Glofitamab is approved for the treatment of adult patients with R/R DLBCL not otherwise specified, DLBCL transformed from indolent follicular lymphoma (FL), or primary mediastinal B-cell lymphoma (PMBCL) after two or more lines of systemic therapy, for those who have previously received or are unable to receive CAR-T cell therapy [11]. Epcoritamab is approved for the same indications and also includes high-grade lymphoma (HGBCL), DLBCL transformed from indolent lymphoma, and grade 3B FL [19,20]. The results of the STARGLO study [21] in 274 patients with RR DLBCL showed an improvement in OS with glofitamab, gemcitabine, and oxaliplatin (Glofit-GemOx) versus rituximab, gemcitabine, and oxaliplatin (R-GemOx) (HR 0.59 [95% CI 0.40–0.89]; *p* = 0.011). Based on these results, the Glofit-GemOx combination may eventually become a treatment option for patients with RR DLBCL. However, given that Glofit-GemOx is not currently available in Canada, it is not further addressed in this paper.

Although potential toxicities of BsAbs also include CRS, ICANS, and cytopenias, they are generally associated with a lower toxicity profile than that seen for CAR-T cell therapy and are more readily available, without the need for bridging or lymphodepleting chemotherapy. The purpose of this paper is to discuss the use of BsAbs in the treatment of R/R DLBCL and to define patient characteristics and logistical factors that may indicate BsAbs as the option of choice in the Canadian treatment landscape.

## 2. Evidence for Bispecific Antibodies

### 2.1. Glofitamab

Glofitamab is an intravenously (IV) administered BsAb with a 2:1 tumor–T-cell binding configuration that confers bivalency for CD20 (B cells) and monovalency for CD3 (T cells), leading to T-cell engagement and redirection to eliminate malignant B cells [3]. Glofitamab is administered over a finite period of 12 cycles, each lasting 21 days [3]. Obinutuzumab is administered prior to the initiation of glofitamab, along with ramp-up dosing in cycle 1 to minimize the risk of CRS.

Glofitamab was evaluated in a pivotal phase II study of 154 patients with R/R DLBCL (including transformed FL, HGBCL, or PMBL) after two or more lines of therapy [3]. At a median follow-up of 13 months, the overall response rate (ORR) was 52% (39% complete response [CR]) and median progression-free survival (PFS) was 5 months. The 12-month PFS and overall survival (OS) rates were 37% and 50%, respectively. The CR rate was 35% for patients who had received prior CAR-T cell therapy versus 42% for those who had not. At a median follow-up of 37.7 months, the ORR was 52% (40% CR) [22]. In patients with a CR, the PFS and OS rates two years after the end of treatment were 57% and 77%, respectively [22].

CRS (any grade) was reported in 63% of patients (grade ≥ 3: 4%), with most events associated with the first three doses [3]. CRS occurring after glofitamab was mainly controlled with corticosteroids and tocilizumab. ICANS was reported in 8% of patients (grade ≥ 3: 3%). Events associated with ICANS (dysphonia, confusional state, and disorientation) were mainly grade 1–2. Infections (any grade: 38%; grade ≥ 3: 15%), neutropenia (any grade: 38%; grade ≥ 3: 27%), anemia (any grade: 31%), and thrombocytopenia (any grade: 25%) were other common adverse events. Long-term data after 3 years of follow-up showed that while B-cell depletion occurred initially in all patients, recovery was observed starting around 18 months after treatment [22].

### 2.2. Epcoritamab

Epcoritamab is a subcutaneously (SC) administered CD3xCD20 T-cell-engaging BsAb that activates T cells, directing them to kill malignant CD20+ B cells, and is continued until treatment failure or intolerance [1]. Ramp-up dosing in cycle 1 is used to minimize the risk of CRS.

A phase I/II pivotal study (EPCORE NHL-1) evaluated epcoritamab in 157 patients with R/R DLBCL (including transformed indolent lymphomas, HGBCL, or PMBL) after two or more lines of therapy [23]. At a median follow-up of 11 months, the ORR was 63%, with 39% of patients achieving a CR. The median PFS was 4 months (not reached [NR] in patients achieving a CR) and the 6-month PFS rate was 44%; median OS was not reached. The ORR was 55% (30% CR) for patients with primary refractory disease and 54% (34% CR) for patients with prior CAR-T cell therapy. The ORR was higher in the subgroup of patients who had not received prior CAR-T cell therapy (69%; 42% CR). After a median follow-up of 37.1 months, the median PFS was 4.2 months (37.3 months in complete responders), and the median OS was 18.5 months (NR in complete responders) [24]. At 36 months, an estimated 63% of complete responders had remained alive. Epcoritamab is given as continuous treatment until disease progression, and further studies are needed to determine the optimal duration of treatment in patients experiencing CR.

Any-grade CRS was reported in 50% of patients with epcoritamab (grade ≥ 3: 3%) [23]. Corticosteroids were given to mitigate CRS during the initial ramp-up. Tocilizumab was used for the management of CRS in 28% of patients. ICANS was reported in 6% of patients (grade ≥ 3: 3%). Common AEs included infections (any grade: 45.2%; grade ≥ 3: 14.6%), neutropenia (any grade: 21.7%; grade ≥ 3: 14.6%), anemia (any grade: 17.8%; grade ≥ 3: 10.2%), and thrombocytopenia (any grade: 13.4%; grade ≥ 3: 5.7%). Pyrexia (any grade: 23.6%), fatigue (any grade: 22.9%), neutropenia (any grade: 21.7%), diarrhea (any grade: 20.4%), and nausea (any grade: 19.7%) were other common adverse events. Long-term data after 3 years of follow-up showed a median decrease in immunoglobulin G levels of around 20% after starting treatment. The highest rate of grade ≥ 3 cytopenias was 27% during the first 8 weeks after starting treatment, with no increase in rates of grade ≥ 3 infections over the 3-year period [24].

### 2.3. Real-World Evidence Studies

A number of real-world evidence (RWE) studies have examined the efficacy and safety of BsAbs for the treatment of DLBCL [25,26,27]. A retrospective study of 209 patients (DLBCL = 155) from 19 U.S. centers demonstrated OR and CR rates of 49% and 23% for epcoritamab, and 53% and 25% for glofitamab, respectively [25]. With a median follow-up of 5 months, the median PFS was 2.7 months (95% CI 2.0, 3.9) and median OS was 7.2 months (95% CI 6.1, NR) for both BsAbs combined. For both BsAbs combined, any-grade CRS occurred in 82 (39.2%) patients and grade ≥ 3 CRS occurred in 9 (4.3%) patients; CRS was treated with tocilizumab in 43 (53.1%) patients. Any-grade ICANS occurred in 24 (11.5%) patients and grade ≥ 3 ICANS occurred in 6 (2.9%) patients; 66.0% of patients with ICANS were treated with corticosteroids. A second retrospective analysis of 70 patients with R/R DLBCL treated with glofitamab within the National Compassionate Use Program in 20 centers across Europe demonstrated an ORR of 44%, with 26% achieving CR [26]. The median PFS was 3.6 months, while the median OS was 5.7 months. All-grade CRS occurred in 40%, with grades 3–4 documented in 2% of patients. Seven (10%) patients presented with ICANS grade 1–2 and one patient (1%) with grade 3. Finally, a multi-center review of 87 (56% DLBCL) adult patients who received epcoritamab, glofitamab, or mosunetuzumab presented safety-related results only, with reporting grade 1, 2, and 3 CRS in 18%, 8%, and 1% of patients, respectively [27]. For the management of CRS, patients required steroids (14%), tocilizumab (10%), and vasopressors (1%). ICANS occurred in six (7%) patients; management included steroids in five (83%) patients.

## 3. Case-Based Discussions

### 3.1. Illustrative Case 1


**Key clinical features**
75-year-old female;Comorbidities include hypertension, type 2 diabetes, and osteoarthritis;Presented with fatigue, night sweats, and bilateral neck fullness;ECOG PS 2;Labs: mild anemia 9.4 g/dL, LDH 420 U/L (ULN 240);PET/CT: lymphadenopathy above and below diaphragm (maximum 14 cm).
**Diagnosis**
Biopsy of cervical lymph node suggests DLBCL, ABC subtype.
**Initial Treatment**
Treated with 6 cycles of dose-reduced R-CHOP (with 1 delay due to infection);Complete response on post-treatment PET/CT;20 months later, developed enlarged cervical nodes;PET/CT and biopsy confirmed recurrent DLBCL.
**Second-Line Treatment**
Not considered to be transplant/CAR-T cell candidate;Treated with polatuzumab–BR for 6 cycles, achieved CR and remained in remission for 18 months.
**Third-Line Treatment**
She now requires further therapy;She is now frailer, with ECOG PS 3, and is being considered for BsAb.


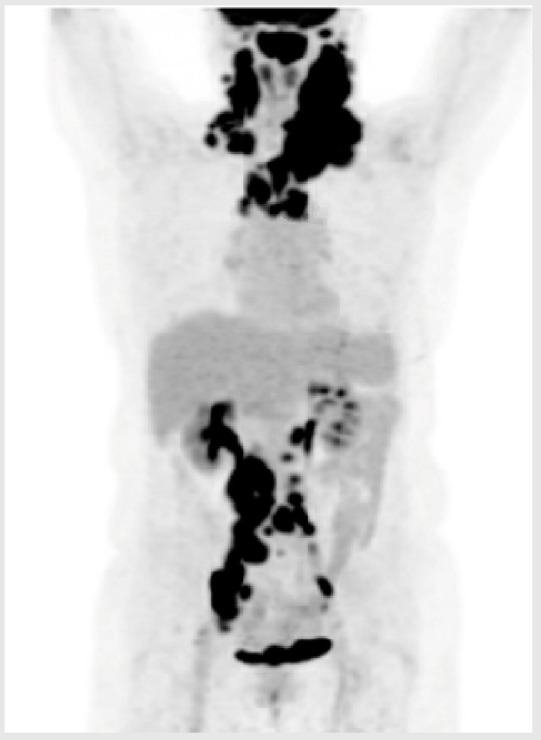



BsAb—bispecific antibody; BR—bendamustine–rituximab; CT—computerized tomography; CR—complete response; DLBCL—diffuse large B-cell lymphoma; ECOG—Eastern Cooperative Oncology Group; LDH—lactate dehydrogenase; PET—positron emission tomography; PS—performance status; ULN—upper limit of normal.

#### Key Points for Illustrative Case 1

Guidelines for CAR-T cell therapy eligibility may vary somewhat between centers, but patients must meet a minimal level of fitness to be considered;Eligibility for CAR-T cell therapy is based on key clinical factors, including adequate cardiac function and a minimum renal function (CrCl greater than 30–45 mL/min). Age and traditional eligibility criteria for HDT-ASCT are less emphasized in favor of general performance status and comorbidities (target ECOG performance score: 0–2);Given the manufacturing time for CAR-T cells, a moderate tumor burden and moderate progression kinetics are necessary to ensure a safe waiting period and treatment trajectory. Rapid disease progression and high tumor burden can compromise general performance status and CAR-T cell efficacy, due to the unpredictable effectiveness of bridging therapies [28];Many patients will need to travel within their province or to another province to receive CAR-T cell therapy, which requires a suitable performance status. For individuals with comorbidities or symptoms due to lymphoma progression that limit their ability to travel, BsAbs may be a more practical and accessible treatment alternative;Additionally, the need for a caregiver can also be a limiting factor, either due to the caregiver’s unavailability or the patient’s feeling of being a burden;Given that BsAbs and CAR-T cell therapy currently target distinct antigens, CD20 and CD19, respectively, a referral for CAR-T cell therapy could be subsequently considered if the patient’s overall condition improves with BsAbs. However, considering this patient’s age of 75 years and her history of multiple comorbidities, the CAR-T cell process may not be feasible and BsAbs may be the preferrable alternative option.

### 3.2. Illustrative Case 2


**Key Clinical Features**
23-year-old male;Presents with drenching night sweats, fever, anasarca, and widespread lymphadenopathy;LDH 1200 U/L (ULN 250);ECOG PS 2, IPI 4.
**Diagnosis**
Groin node biopsy: T-cell/histiocyte-rich large B-cell lymphoma;Stage IVB, bone marrow and liver involved.
**Initial Treatment**
Given RCHOP × 5 cycles;Primary refractory, with recurrent fevers, and increasing LDH.
**Second-Line Treatment**
R-GDP: cycle 1 received;Initial improvement, but recurrent fevers and increasing LDH prior to cycle 2;PET confirmed progression.
**Third-Line Treatment**
Plan for CAR-T cell therapy;Time from CAR-T cell therapy consultation to leukapheresis was 4 days;No holding therapy administered, but bridging therapy with Pola-R considered;Unfortunately, T-cell collection was insufficient to enable adequate CAR-T cell product;Potential candidate for BsAbs.


#### Key Points for Illustrative Case 2

Predictive factors for CAR-T manufacturing failure include a low CD3+ T-cell count (<150–300/µL), low proportions of naïve (CD45RA+) and central memory (CCR7+) T cells, high monocyte contamination (>40% CD14+ cells), and a suboptimal CD4/CD8 ratio (<1:3). Extensive chemotherapy, particularly with agents like bendamustine, reduces T-cell functionality and availability, while cumulative treatments and disease-related T-cell exhaustion further impair success. A high tumor burden (bulk disease) is also associated with reduced CAR-T manufacturing efficiency and outcomes [29,30]. T-cell fitness is an important component for the optimization of immunotherapeutic approaches, including CAR-T cell therapy and BsAbs [31]. Unlike CAR-T cell therapy, BsAbs involve repeated dosing with intervals between treatments, which may allow for newly regenerated T cells to contribute to the therapeutic process [32]. Moreover, unlike with CAR-T cell therapy, bendamustine-containing regimens prior to BsAbs do not appear to impact outcomes, although more data are required to support this [33];Patients with rapidly progressing refractory lymphoma are often excluded from clinical trials, as their disease progression does not allow for the screening period required for enrollment. Although bridging therapies can be attempted, they are often ineffective, prohibiting patients from proceeding to CAR-T cell therapy;Even though BsAbs have a delayed onset of action due to the ramp-up phase to mitigate the risk of CRS, the timeline associated with a CAR-T cell therapy trajectory remains longer. This is especially important here, where the disease is aggressive;If rapid disease progression does not allow for CAR-T cell therapy, initiating a BsAb may be a consideration. However, real-world evidence on the efficacy of BsAbs in this setting is awaited, as well as data on the use of BsAbs as a possible bridging therapy;Importantly, data suggest that CAR-T cell therapy remains effective in patients with R/R LBCL after prior exposure to BsAbs, suggesting that the administration of a BsAb does not preclude patients from receiving future CAR-T cell therapy [34].

### 3.3. Illustrative Case 3


**Key Clinical Features**
46-year-old Indigenous male from remote area of northern Canada;Comorbidities: HTN, CAD, osteoarthritis;Presented with large neck mass;Stage IV, IPI 4/5, ECOG PS 2.
**Diagnosis**
Sent to treatment center 2000 km away for cervical node core biopsy, non-diagnostic;Sent back again for excisional biopsy, interval between biopsies was 2 months;DLBCL, non-GCB, double expressor, no *MYC* rearrangement.
**Initial Treatment**
Plan for R-CHOP x 6 but patient did not show for 1st cycle;Treatment start was delayed 6 weeks and patient elected to return home between cycles;PET scan after 3 cycles demonstrates mixed response.
**Second-Line Treatment**
Plan for ASCT and discussed salvage with R-ICE versus R-GDP;Chose R-ICE to have longer time at home between treatment;Progression after 2 cycles.
**Third-Line Treatment**
Discussion of CAR-T cell therapy versus BsAb therapy, patient elected to proceed with BsAb therapy.


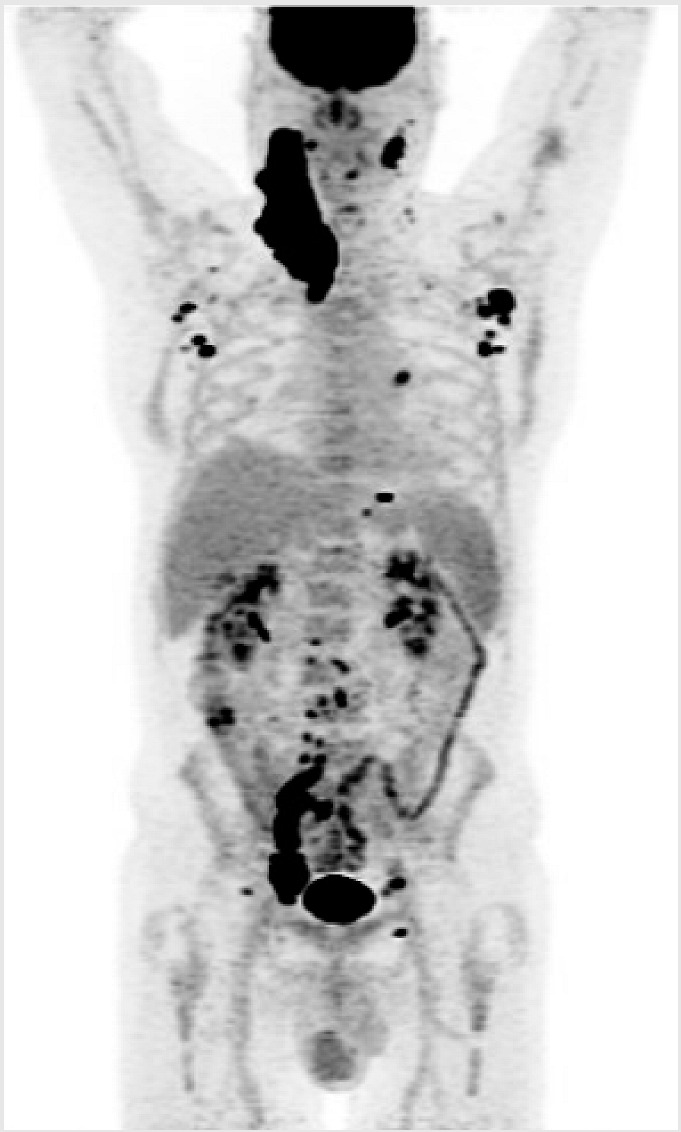



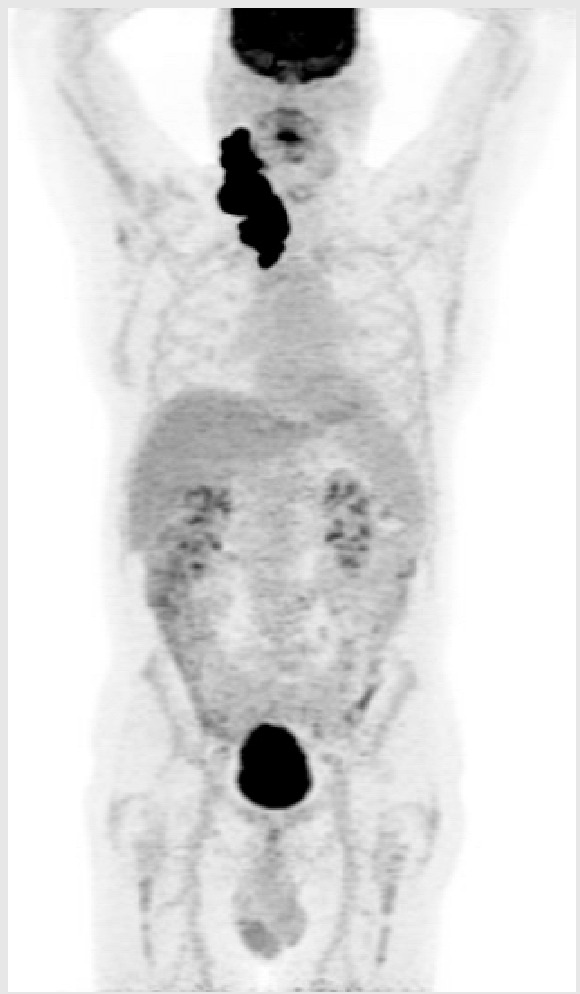



ASCT—autologous stem cell therapy; BMT—bone marrow transplant; BR—bendamustine–rituximab; CAD—coronary artery disease; CR—complete response; DLBCL—diffuse large B-cell lymphoma; ECOG—Eastern Cooperative Oncology Group; HDT—high-dose therapy; HTN—hypertension; IPI—international prognostic index; LDH—lactate dehydrogenase; MI—myocardial infarction; PET—positron emission tomography; PR—partial response; PS—performance status; R-CHOP—rituximab–cyclophosphamide–doxorubicin–vincristine–prednisone; R-GDP—rituximab–gemcitabine–cisplatin–dexamethasone; ULN—upper limit of normal.

#### Key Points for Illustrative Case 3

CAR-T cell therapy requires a coordinated process involving significant contributions by the patient and healthcare team [14,17]. CAR-T cell therapy requires, at minimum, the following: an initial consultation; leukapheresis; and a lymphodepleting therapy and infusion with the necessity of the patient staying in proximity to the CAR-T cell center to complete a total of one month after infusion. The vast majority of patients are hospitalized during the initial 2 weeks after CAR-T infusion. It is also recommended that patients have a caregiver accompany them during the CAR-T cell therapy process;The administration of BsAbs involves frequent visits for therapy, which contrasts with the one-time administration of the CAR-T cell product. Individual preferences will likely influence patient decision-making;The length of time away from home was the driving factor in this patient’s preference for BsAbs. Despite the frequency of administration of BsAb therapy, he preferred to travel back and forth for the shorter visits;For many patients, the necessity of travel and an extended stay near the CAR-T cell center is a meaningful barrier. The requirement to travel for treatment involves personal, familial, financial, and professional considerations. Patients’ geographic distance to treatment centers is a major limitation for many patients, with those residing 2–4 h away being 40% less likely to access CAR-T cell therapy [35]. The mapping of CAR-T cell therapy administered in the province of Quebec illustrated this unfortunate reality [36];Some remote centers may initiate the ramp-up of BsAbs in a regional hospital or cancer center, with later cycles administered closer to home in an infusion clinic;The incidence of CRS occurrence from cycle 2 onwards is very low (less than 5%), and its severity is mild (typically grade 1 or 2) [1,3]. Moreover, the occurrence of CRS from cycle 2 onwards is often observed in patients who experience more severe or prolonged CRS during cycle 1, making it more predictable. The incidence and severity of neurotoxicity with BsAbs is also low, further supporting the feasibility of administration at local centers with regional oversight;Having a caregiver present during the ramp-up of BsAbs is recommended but not absolutely necessary, as long as patients are reliable and compliant. If there are particular concerns about a patient’s condition and reliability, their monitoring period in the hospital could be extended;For patients who do not want to or cannot travel for CAR-T cell therapy, BsAbs may represent a valuable treatment option. However, the duration of follow-up from BsAb clinical trials remains insufficient to assess the curative potential of this approach.

### 3.4. Illustrative Case 4


**Key clinical features**
58 -year-old male;No comorbidities;Presented with abdominal discomfort;ECOG PS 1;Labs: LDH 350 U/L (ULN 240);PET/CT: paravertebral soft tissue mass at T7 with extension into right lower lobe, right pelvic sidewall mass (maximum 6 cm).
**Diagnosis**
Core biopsy of abdominal mass: DLBCL, GCB subtype, no *MYC* rearrangement.
**Initial Treatment**
Treated with 6 cycles of R-CHOP;CR on post-treatment PET/CT;14 months later, developed recurrent abdominal pain, PET/CT and biopsy confirmed recurrent DLBCL.
**Second-Line Treatment**
Planned for salvage and ASCT, but had progression after 2 cycles of R-GDP.
**Third-Line Treatment**
Referred for CAR-T cell therapy;Received 1 cycle of Pola-R bridging followed by axicabtagene ciloleucel;CR on PET/CT at 3 months;Progression on PET/CT at 6 months post CAR-T cell therapy.
**Fourth-Line Treatment**
He has recently been given epcoritamab.


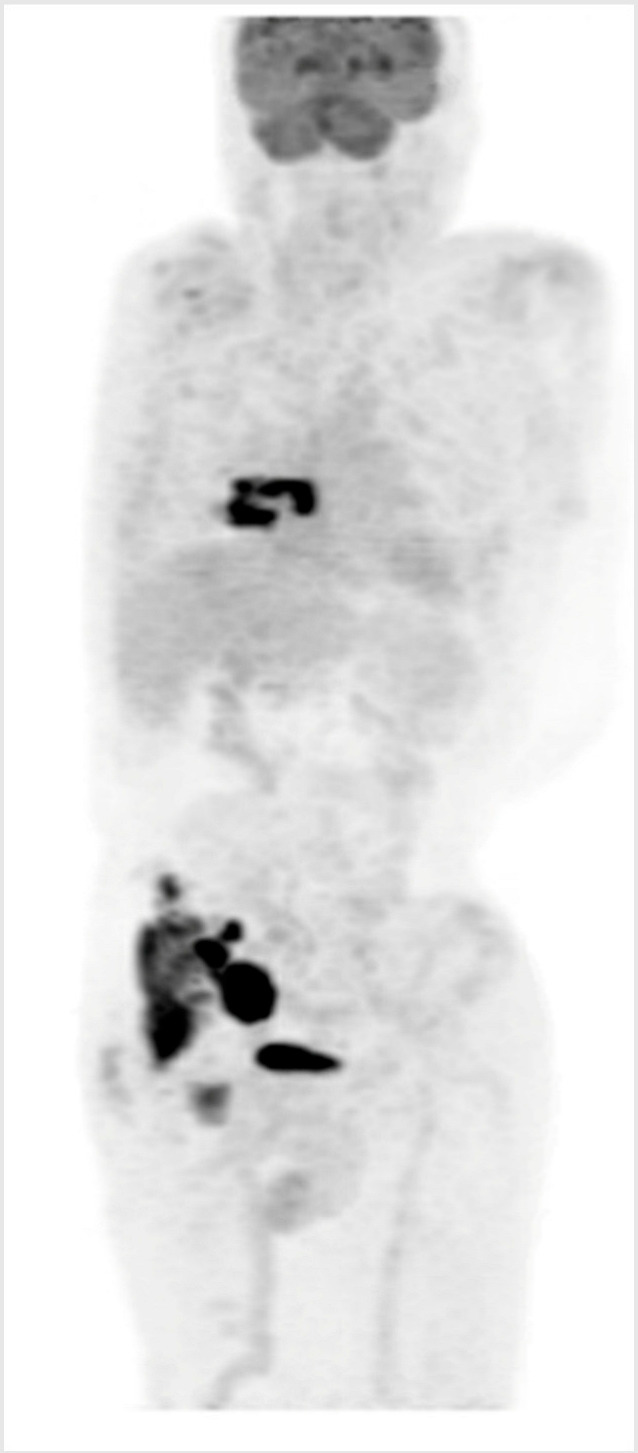



ASCT—autologous stem cell transplant; BsAB—bispecific antibody; CT—computerized tomography; CR—complete response; DLBCL—diffuse large B-cell lymphoma; ECOG—Eastern Cooperative Oncology Group; LDH—lactate dehydrogenase; PET—positron emission tomography; PS—performance status; R-CHOP—rituximab–cyclophosphamide–doxorubicin–vincristine–prednisone; R-GDP—rituximab–gemcitabine–dexamethasone; ULN, upper limit of normal.

#### Key Points for Illustrative Case 4

Unfortunately, this otherwise healthy man exhibits chemo-refractory disease after an initial benefit from R-CHOP and did not sustain a durable benefit from CAR-T cell therapy;BsAbs have demonstrated efficacy for patients with R/R DLBCL regardless of prior exposure to CAR-T cell therapy [1,3]. This case illustrates the potential use of BsAbs following CAR-T cell therapy failure, with CAR-T cell therapy initially prioritized due to its longer available follow-up and known curative potential.

## 4. Canadian Perspective

CAR-T cell therapy is currently offered across Canada in twelve centers, and is available in seven out of ten provinces, but not within the three territories [37]. Access to CAR-T cell therapy is further limited by patient ineligibility due to poor performance status, comorbidities, or a rapidly progressing burden of disease; ineffective bridging therapies; manufacturing limitations; and limited healthcare resources [1]. BsAbs generally have a more favorable toxicity profile and are available for immediate use, and can be administered more broadly, making them a more accessible treatment option for certain patient populations (Table 1). However, long-term follow-up evaluating the curative potential of BsAb therapy is not yet available; thus, in settings where CAR-T cell therapy is feasible, it should remain the initial consideration.

### 4.1. BsAbs in Patients Ineligible for CAR-T Cell Therapy

Patients may be ineligible for CAR-T cell therapy as a result of frailty, organ dysfunction, inadequate performance status, or aggressive disease needing immediate treatment. In cases where performance status is suboptimal, such as in Case 1, BsAbs may be a desirable option, as the toxicity profile of BsAbs is generally more favorable than that of CAR-T cell therapy. However, there is currently no evidence from clinical trials to support this strategy. Importantly, the administration of a BsAb does not preclude patients from receiving future CAR-T cell therapy if they become eligible due to improved functional status. A recent retrospective analysis of patients with R/R large B-cell lymphoma (LBCL) treated with CD19-targeted CAR-T cells after prior BsAb exposure demonstrated a best ORR of 85% (43% CR), without significant differences in patients who had not previously responded to a BsAb [34]. At a median follow-up of 10.5 months, the median PFS was 6.6 months (95% CI 2.6-NR) and median OS was not reached (95% CI 9.0-NR). The second part of the same study included a matched control group of BsAb-naïve patients treated with CAR-T cell therapy. The BsAb-exposed group achieved a higher ORR compared with the control group (86% vs. 55%, *p* = 0.02), but CR, 1-year PFS, and 1-year OS were not statistically different between groups. Data suggest that CAR-T cell therapy remains effective in patients with R/R LBCL after prior exposure to BsAbs.

Patients with rapidly progressing disease may be ineligible for CAR-T cell therapy due to their inability to wait for the apheresis and manufacturing process, as in Case 2. Systemic bridging therapies, radiation therapy, or other intervention strategies may enable some patients with rapid disease progression to wait through the manufacturing period and proceed to CAR-T cell therapy. However, bridging therapies have limited efficacy and can be associated with notable toxicities, such as cytopenias [12,13]. The required ramp-up period for BsAbs can also delay anti-tumor benefit, although accelerated ramp-up schedules of BsAbs for high-burden disease have been explored [38,39]. Administering radiotherapy to critical sites of disease may enable the effective use of BsAbs for some patients with rapidly evolving disease [40].

### 4.2. BsAbs as an Alternative to CAR-T Cell Therapy Based on Patient Preference

For patients who are unable or unwilling to travel for CAR-T cell therapy, as in Case 3, BsAbs may provide a more feasible treatment option. BsAbs would ideally be administered locally at treatment centers or infusion clinics in remote areas. However, currently, most patients are required to travel to regional clinics or cancer centers for treatment initiation due to the risk of CRS, which occurs primarily during cycle 1. Patients also require brief hospitalization (~24–48 h) during the highest risk period of CRS, which requires some patients to be managed at specialized centers.

### 4.3. Potential of BsAbs for Bridging to CAR-T Cell Therapy

Bridging therapy refers to treatment that is delivered after apheresis and before CAR-T cell therapy infusion. Bridging therapies must be rapidly effective and minimally toxic with a short washout period. There are limited data evaluating the use of BsAbs as bridging therapy, and currently, BsAbs are not funded in Canada for this indication. The median time to response with epcoritamab is 1.4 months, and the median time to CR is 1.4 months with glofitamab, corresponding to the timing of the first imaging performed in clinical trials [23,41]. As CAR-T cell therapy manufacturing is typically completed in three weeks, the timeline of BsAbs (including the need for ramp-up in cycle 1) may be insufficient to provide timely disease control as a bridging therapy. It remains to be seen whether BsAbs could be used as bridging therapy to CAR-T cell therapy in DLBCL and whether the ramp-up period could be reduced in these cases to allow for faster response. Currently, radiation therapy, polatuzumab vedotin with rituximab, conventional chemotherapy, and steroids are commonly used as bridging therapies in Canada. Combinations of BsAbs with chemotherapy may circumvent this limitation and offer a novel bridging strategy [21].

### 4.4. BsAbs Following CAR-T Cell Therapy

BsAbs are the preferred treatment of choice for progression following CAR-T cell therapy, based on their demonstrated efficacy and favorable toxicity profile. As shown in Case 4, patients failing CAR-T cell therapy are ideal candidates for BsAbs. The CR rate in patients with prior CAR-T cell therapy is 36% with epcoritamab and 37% with glofitamab [1]. In addition, the median duration of CR is 36.1 months for epcoritamab [24] and 29.8 months for glofitamab [22] following CAR-T cell therapy. Moreover, in a real-world multi-center cohort of 64 patients treated with BsAbs after CAR-T cell failure, the ORR was 54% (29/54) with a CR rate of 33% (18/54). At a median follow-up of 400 days after the initiation of BsAbs, 66.7% patients remained in CR [42]. Polatuzumab–BR is an alternative systemic option available in Canada after CAR-T cell therapy. However, response may be less durable, and tolerance may be limited due to the risk of cytopenias [1,41,43]. Radiation therapy may be a consideration for localized disease in selected patients.

## 5. Conclusions

CAR-T cell therapy has revolutionized outcomes for patients with R/R large B-cell lymphoma; however, challenges related to access, efficacy, and toxicity remain barriers to its use. BsAbs may address these challenges for many patients, as they have become the preferred therapy after CAR-T cell therapy failure and provide an effective option for patients who are ineligible for CAR-T cell therapy. Ongoing studies of BsAb combinations may further improve outcomes for patients with R/R DLBCL.

## Figures and Tables

**Table 1 curroncol-32-00142-t001:** Patients to consider for bispecifics.

Patient Type	Patient Description	Benefits of Bispecifics
**Ineligible for CAR-T cell therapy**	Inadequate performance status Organ dysfunction	Provide effective therapy with durable benefit and favorable toxicity profile Incidence and severity of toxicities lower than CAR-T cell therapy
**Eligible for CAR-T cell therapy**	Rapidly progressing disease Borderline PS for CAR-T cell therapy Unable/unwilling to travel for CAR-T cell therapy Concern about CAR-T cell therapy toxicity profile	Timely available therapy (although ramp-up may be associated with delayed response) Provide more flexible option and offered in more centers than CAR-T cell therapy, requiring less travel Do not preclude future CAR-T cell therapy
**Post CAR-T cell therapy**	All fit patients	Preferred option in terms of efficacy and safety, compared to available alternatives Post-ramp-up ease of administration Clinical trials should also be considered, but availability often limited

BR—bendamustine–rituximab.

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
