# Peer review of "Optimal Use of Bispecific Antibodies for the Treatment of Diffuse Large B-Cell Lymphoma in Canada"

_curroncol, 2025, doi:10.3390/curroncol32030142_

Round 1
Reviewer 1 Report
Comments and Suggestions for Authors
In this paper, the authors illustrate and comment four refractory/relapsed LBCL cases commonly occurring in clinical practice where BsAb therapy can be considered: in CAR-T ineligible patients; in CAR-T eligible patients but not proceeding to CAR-T due to disease progression, lymphocyte collection failure, or patient’s choice; in post-CAR-T settings. Canadian prospectives about the use of BsAb in LBCL are finally presented
The paper is interesting, scientifically and methodologically accurate, complete and persuasive.
The reference section is adequate.
Author Response
No responses needed - we thank you for your review.
Reviewer 2 Report
Comments and Suggestions for Authors Major comments: - Real-world experience data on BsAbs in DLBCL should bez added to Part 2 „Evidence for BsAb“ to characterise better the benefit of BsAb in real-world clinical practice – e.g. Brooks TR et al ASH 2024 abstract O111 or Wurm-Kuczera R et al. EHA 2024 abstract P1166 - Given the expected approval of Gloftamab-GemOx combination in 2+L treatment of RR DLBCL based on STARGLO trial results as early as 2025, this combination should be mentioned in the article as near future practice - The article emphasises the risks associated with CART therapy in several parts, the long-term risks associated with BsAb treatment such as hypoglobulinemia and risk of infection should also be highlighted - starting with reference 28, references are missing in reference list Minor comments: 26-27 - if the Pola-R-CHP regimen is available in Canada as a standard option for DLBCL IPI (2)3-5 patients , similar to the EU and US region, this regimen should be mentioned alongside R-CHOP 36 - inaccurate statement - text does not apply to all CART products in general, only axi-cel and liso-cel showed benefit in 2nd line setting, not tisa-cel (BELINDA trial). Citation to TRANSFORM study should be added 43-45 - Although the above statement is generally acceptable, in general, CART toxicity is not necessarily expressed in all patients and also depends on the specific product used (axi-cel vs. liso-cel). Consider the change of the wording to "might be associated with significant toxicity". 50 - add a citation to the statement 253-254 - I understand the writers point of statement, however, no data from clinical trials are available to support this statement as both registration trials for glifitamab and epcoritamab included only patients with ECOG 0-1 and 0-2 respectively. Real world evidence date suggests poor oucomes in this setting (see Brooks TR, ASH 2024 abstract O111) 285 – at present, when axi-cel and liso-cel are typicallyused in the second line in most patients, this approach is not applicable - BsAbs are registered for use in 3+ lines 294 – reference to MM procedures is rather irrelevant for high grade B-lymphomas 302 – inaccurate statement - bendamustine may be used after cell apheresis, it is not suitable only before mononuclear cell apheresisAuthor Response
Thank you for your thoughtful comments, please find our responses below:
Major comments
1. Real-world experience data on BsAbs in DLBCL should be added to Part 2 „Evidence for BsAb“ to characterise better the benefit of BsAb in real-world clinical practice – e.g. Brooks TR et al ASH 2024 abstract O111 or Wurm-Kuczera R et al. EHA 2024 abstract P1166
We agree with the reviewer’s comment and have added a section on real-world evidence studies on lines 123-139:
“Real-World Evidence Studies
A number of real-world evidence (RWE) studies have examined the efficacy and safety of BsAbs for the treatment of DLBCL. 1,2,3 A retrospective study of 209 patients (DLBCL = 155) from 19 U.S. centers demonstrated ORR and CR rates of 49% and 23% for epcoritamab, and 53% and 25% for glofitamab, respectively. 1 With a median follow-up of 5 months, median PFS was 2.7 months (95% CI 2.0, 3.9) and median OS was 7.2 months (95% CI 6.1, NR), for both BsAbs combined. For both BsAbs combined, any-grade CRS occurred in 82 (39.2%) patients and grade ≥3 CRS occurred in 9 (4.3%) patients; CRS was treated with tocilizumab in 43 (53.1%) patients. Any grade ICANS occurred in 24 (11.5%) patients and grade ≥3 ICANS occurred in 6 (2.9%) patients; 66.0% of patients with ICANS were treated with corticosteroids. A second retrospective analysis of 70 patients with R/R DLBCL treated with glofitamab within the National Compassionate Use Program in 20 centers across Europe demonstrated an ORR of 44%, with 26% achieving CR. 2 The median PFS was 3.6 months, while the median OS was 5.7 months. All-grade CRS was 40%, with grades 3-4 documented in 2% of patients. Seven (10%) patients presented with ICANS grade 1-2 and one patient (1%) with grade 3. Finally, a multicenter review of 87 (56% DLBCL) adult patients who received epcoritamab, glofitamab, or mosunetuzumab presented safety results only, reporting Grade 1, 2, and 3 CRS in 18%, 8%, and 1% of patients, respectively. 3 For the management of CRS, patients required steroids (14%), tocilizumab (10%), and vasopressors (1%). ICANS occurred in 6 (7%) patients; management included steroids in 5 (83%) patients.”
2. Given the expected approval of Gloftamab-GemOx combination in 2+L treatment of RR DLBCL based on STARGLO trial results as early as 2025, this combination should be mentioned in the article as near future practice
We agree with the reviewer that the Glofit-GemOx combination should be mentioned in the paper and have added in lines 61-65. However, because Glofit-GemOx is not yet available in Canada, the authors feel it is beyond the scope of this paper to discuss further.
“Results of the STARGLO study4 in 274 patients with RR DLBCL showed an improvement in OS with glofitamab, gemcitabine, oxaliplatin (Glofit-GemOx) versus rituximab, gemcitabine, oxaliplatin (R-GemOx) (HR 0.59 [95% CI 0·40–0·89]; p=0·011). Based on these results, the Glofit-GemOx combination may eventually become a treatment option for patients with RR DLBCL. However, given that Glofit-GemOx is not currently available in Canada, it is not further addressed in this paper.”
3. The article emphasises the risks associated with CART therapy in several parts, the long-term risks associated with BsAb treatment such as hypoglobulinemia and risk of infection should also be highlighted
We acknowledge the reviewer’s point about long-term hyperglobulinemia and potential risk of infections. Therefore, we have added in the following details to the paper:
Lines 94-96: “Long-term data after 3 years of follow-up showed that while B-cell depletion occurred initially in all patients, recovery was observed starting around 18 months after treatment.”
Lines 119-121: “Long-term data after 3 years of follow-up showed a median decrease in immunoglobulin G levels of around 20% after starting treatment. The highest rate of grade ≥3 cytopenias was 27% during first 8 weeks after starting treatment, with no increase in rates of grade ≥3 infections over the 3-year period.”
4. Starting with reference 28, references are missing in reference list
We agree with the reviewer and have added in the missing references.
Minor comments
1. Line 26-27: If the Pola-R-CHP regimen is available in Canada as a standard option for DLBCL IPI (2)3-5 patients, similar to the EU and US region, this regimen should be mentioned alongside R-CHOP
We acknowledge the reviewer’s point, but unfortunately the Pola-R-CHP regimen is not available in Canada.
2. Line 36: inaccurate statement - text does not apply to all CART products in general, only axi-cel and liso-cel showed benefit in 2nd line setting, not tisa-cel (BELINDA trial). Citation to TRANSFORM study should be added
Revised to: "CAR-T cell therapy has emerged as a superior option to platinum-based salvage and HDT-ASCT for patients with refractory or early relapsing disease, with axicabtagene ciloleucel and lisocabtagene maraleucel demonstrating significant reductions in both relapse risk and mortality in the second-line setting.5, 6 Axicabtagene ciloleucel is now the standard-of-care in Canada for patients with refractory disease or who relapse within one year of completion of frontline treatment, with lisocabtagene maraleucel not funded yet in Canada. 7 The CAR-T cell therapies, axicabtagene ciloleucel8, lisocabtagene maraleucel9, and tisagenlecleucel10 have also shown promising efficacy in the third-line setting and represent the preferred treatment option in Canada for suitable patients who have not previously received it in the second-line setting.7"
3. Line 43-45: Although the above statement is generally acceptable, in general, CART toxicity is not necessarily expressed in all patients and also depends on the specific product used (axi-cel vs. liso-cel). Consider the change of the wording to "might be associated with significant toxicity".
We agree with the reviewer and have modified the statement as follows:
“However, CAR-T cell therapy may be associated with notable toxicities, such as cytokine release syndrome (CRS) and immune effector cell–associated neurotoxicity syndrome (ICANS), as well as severe and prolonged cytopenias. 11,12
Access to CAR-T cell therapy remains a challenge due to logistical, geographical, and resource-related constraints. 13 Furthermore, eligibility for CAR-T cell therapy is often restricted due to patient comorbidities, potential CAR-T cell related toxicities, and…”
4. Line 50: Add a citation to the statement
We agree and have added in a citation to support the statement below:
“Up to one-third of patients referred for CAR-T cell therapy across Canada do not proceed, primarily due to prohibitive lymphoma progression.”
Reference: Steven Shi , Eva Laverdure , Sandra R. A. Cohen , Olivier Veilleux Real-world barriers to CAR-T access: A Canadian referral center perspective.: ASCO Annual Meeting; 2024.
5. Line 253-254: I understand the writers point of statement, however, no data from clinical trials are available to support this statement as both registration trials for glifitamab and epcoritamab included only patients with ECOG 0-1 and 0-2 respectively. Real world evidence date suggests poor outcomes in this setting (see Brooks TR, ASH 2024 abstract O111)
We acknowledge the reviewer’s point about the lack of clinical evidence to support the use of BsAbs in patients with suboptimal performance status. However, given the more tolerable safety profile for BsAbs and the fact that ECOG alone is not the best marker to decide fitness for therapy, we have left in this point with a modification to acknowledge the lack of clinical trial data:
“In cases where performance status is suboptimal, such as in Case 1, BsAbs may be a desirable option, as the toxicity profile of BsAbs is generally more favorable than CAR T-cell therapy. However, there is currently no evidence from clinical trials to support this strategy, since most clinical trials included patients with an ECOG score of 0-1.”
6. Line 286: At present, when axi-cel and liso-cel are typically used in the second line in most patients, this approach is not applicable. BsAbs are registered for use in 3+ lines
While we agree that BsAbs are not currently used as bridging therapy, the authors have included this section as a potential for future consideration. We have modified the section title to make this clearer to: “Potential of BsAbs for Bridging Therapy”. We do acknowledge that this is not a current indication of BsAbs: “There is limited data evaluating the use of BsAbs as bridging therapy, and currently BsAbs are not funded in Canada for this indication”.
7. Line 262: Reference to MM procedures is rather irrelevant for high grade B-lymphomas
We agree with the reviewer and have removed reference to MM procedures (content below):
“Some limited data exists in multiple myeloma (MM), where BsAbs have shown to be a potent and safe option as bridging therapy, achieving the highest ORR (100%) compared with chemotherapy, anti-CD38, or anti-SLAMF7 antibody–based regimens (46%).14 ”
8. Line 267: inaccurate statement - bendamustine may be used after cell apheresis, it is not suitable only before mononuclear cell apheresis
The reviewer is correct. We have removed the sentence below:
“Avoidance of highly lymphodepleting agents, such as bendamustine is recommended, as it may compromise outcomes following CAR T-cell therapy. 15”
Reviewer 3 Report
Comments and Suggestions for Authors
This is an excellent review of the use of bispecific antibodies for the treatment of lymphoma in Canada detailing several clinical cases to illustrate the main concerns with this type of therapy and its use in both the urban and rural settings in a large country. I believe that the paper will be very instructive to the readership and found it both interesting and thought provoking in scope. The treatment of lymphomas has been a great success for oncology but he implementation of newer treatments presents challenges in decision making as detailed in this review. The authors have presented an excellent discussion and I was pleased with the format and information provided.
Author Response

(The authors gave the same response as above.)

Reviewer 4 Report
Comments and Suggestions for Authors
This is an excellent manuscript on use of bispecific antibodies in the treatment of r/r DLBCL. The authors present the barriers to CAR-T cell therapies and post CAR-T relapse setting and through the interest cases lead the reader through various settings of when epcoritumab and glofitamab may be used to fill in this unmet medical need. Furthermore, it is evidence based with authors providing strong rationale for their use. I do not see any need for revision.
Author Response

(The authors gave the same response as above.)
